# Impact of Delayed Intravitreal Anti-Vascular Endothelial Growth Factor (VEGF) Therapy Due to the Coronavirus Disease Pandemic on the Prognosis of Patients with Neovascular Age-Related Macular Degeneration

**DOI:** 10.3390/jcm11092321

**Published:** 2022-04-21

**Authors:** Jae-Gon Kim, Yu Cheol Kim, Kyung Tae Kang

**Affiliations:** Department of Ophthalmology, Keimyung University Dongsan Hospital, Keimyung University School of Medicine, Daegu 42601, Korea; jgkim9418@naver.com (J.-G.K.); eyedr@dsmc.or.kr (Y.C.K.)

**Keywords:** COVID-19, retina, neovascular AMD, intravitreal VEGF injection

## Abstract

This study estimated the outcome of delayed intravitreal anti-vascular endothelial growth factor (VEGF) therapy due to the coronavirus (COVID-19) disease pandemic on the prognosis of patients with neovascular age-related macular degeneration (nAMD). This study retrospectively enrolled 57 nAMD patients whose intravitreal anti-VEGF injections were delayed for >2 weeks between February and June 2020. Best-corrected visual acuity (BCVA), central subfield thickness (CST), and anatomical characteristics were evaluated before (baseline), on the day, and at 2, 4, and 6 months after the delayed injection, and risk factors were identified. The average injection interval before and after treatment delay was 3.05 ± 1.45 and 2.41 ± 1.46 months, respectively (*p* = 0.002). The CST at baseline and on the day of delayed injection was 227.82 ± 62.46 and 267.26 ± 77.74 µm, respectively (*p* < 0.001). The average BCVA decreased from 0.29 ± 0.29 logMAR (baseline) to 0.38 ± 0.31 logMAR (6 months) (*p* = 0.001). The maximum subretinal fluid (SRF) height increased from 84.32 ± 89.33 µm (baseline) to 121.38 ± 103.36 µm (6 months) (*p* = 0.027). A higher baseline maximum SRF height was associated with less SRF height deterioration 6 months later (*p* < 0.001). Delayed intravitreal anti-VEGF therapy caused by the COVID-19 pandemic has worsened BCVA and residual SRF in nAMD patients after a temporary recovery. The baseline SRF reduce the degree of SRF height deterioration.

## 1. Introduction

Age-related macular degeneration (AMD) is a major cause of blindness worldwide, accounting for 6–9% of the causes of blindness. It is a degenerative disease caused by ageing of the human retina, and its pathology mainly occurs in the macula lutea. AMD is associated with complex interactions between ageing, genetic contributions, and environmental factors. In later stages, AMD can progress to neovascular AMD (nAMD), which causes >90% of severe vision loss that may occur in AMD, as untreated neovascularization usually leads to almost inevitable extensive macular damage and irreversible vision loss [1,2]. nAMD is characterized by choroidal neovascularization with intraretinal or subretinal leakage, hemorrhage, and detachment of the retinal pigment epithelium (RPE). Despite advances in treatment methods, none of the treatments currently used for nAMD can cure the disease or reverse the process. Injecting anti-VEGF drugs, such as aflibercept, ranibizumab, and bevacizumab, into the vitreous cavity is currently the gold standard for nAMD. This type of treatment is known as intravitreal anti-vascular endothelial growth factor (VEGF) injection [3,4].

During the coronavirus disease (COVID-19) pandemic, concerns about the spread of infection reduced the number of outpatient visits, leading to a decrease in intravitreal injections. Between February and March 2020, the first COVID-19 outbreak in South Korea was detected in Daegu, a city with a population of nearly 2.5 million. At the end of the first outbreak, the number of positive COVID-19 cases was >5000, because of which, people avoided even necessary activities, such as hospital visits. Additionally, healthcare providers postponed scheduled preventive care appointments to reduce the spread of infection, and outpatient evaluations were limited to more urgent care. This resulted in poor adherence to nAMD treatment, a problem that was more remarkable among older patients [5]. If the ocular condition was stable, treatment over the phone was recommended rather than a direct visit to a medical institution [6]. Because AMD occurs in the elderly population, receiving care over the phone may not be appropriate and effective. The COVID-19 pandemic has negatively influenced the diagnosis and treatment of diseases. COVID-19 has significantly delayed the treatment of patients with nAMD, an issue that has been significantly related to the deterioration of short-term outcomes in these patients [7,8]. COVID-19 has also delayed the diagnosis of new AMD cases, demonstrating that patients with newly diagnosed exudative nAMD had worse clinical characteristics at presentation and short-term visual outcomes [9]. Whatever the reason, any delay in diagnosis or treatment, such as intravitreal anti-VEGF injection, can lead to irreversible visual impairment in patients with nAMD [10,11].

In previous studies, the best-corrected visual acuity (BCVA) was consistently reported to worsen, to varying degrees, in nAMD patients whose injection was delayed due to the COVID-19 pandemic [8,12,13,14,15,16], and disease activity showed a worsening [12,15] or recovery. Refs. [8,16] However, the available research on changes in visual acuity and anatomical status in nAMD patients and the analyses of prognostic factors after treatment delay due to COVID-19 is limited [7,17].

This study aimed to compare the visual acuity and anatomical changes in nAMD patients before and 6 months after receiving a delayed intravitreal anti-VEGF injection due to the COVID-19 pandemic. It also aimed to investigate the factors influencing these changes. Therefore, we retrospectively reviewed nAMD patients whose injection was delayed due to COVID-19, checked the changes in the BCVA and OCT parameters over a period of 6 months, and analyzed whether the changes were related to clinical symptoms. Our findings may help to determine the prognosis of nAMD patients whose treatment is delayed due to COVID-19, and prevent similar situations in the future.

## 2. Materials and Methods

This study was approved by the Keimyung University Dongsan Hospital Institutional Review Board on 26 April 2021 (approval number: DSMC 2021-04-060), and it adhered to the tenets of the Declaration of Helsinki. This was a retrospective study of nAMD patients whose intravitreal anti-VEGF injection treatment was delayed for at least 2 weeks because they were not able to attend their scheduled visits at Keimyung University Dongsan Hospital between February and June 2020, a period when the incidence of COVID-19 had increased rapidly in Daegu, Korea. In our hospital, anti-VEGF injection was administered on the day nAMD was diagnosed if the patient met the indication. Ref. [3] Even before the onset of the COVID-19 pandemic, there were some cases where patient visits were delayed due to various reasons, leading to delays in injection. However, only the patients who visited the hospital on the appointment date and received injections on time until the COVID-19 pandemic were included in this study. As this was a retrospective study of medical records, informed consent was waived and not required. Patients who did not require intravitreal anti-VEGF injection or had factors that could affect retinal structures were excluded. The inclusion and exclusion criteria are summarized in the Appendix A. Only one eye of each patient was enrolled in the study. When a patient had nAMD in both eyes, the eye that was most recently treated with anti-VEGF injections for nAMD was considered.

We retrospectively reviewed the patients’ medical records on the last visit before the date of delayed injection (baseline); on the day of the delayed injection; and at 2, 4, and 6 months after the delay. The BCVA as the logarithm of the minimum angle of resolution (logMAR) for each visit was recorded. All patients underwent swept-source optical coherence tomography (SS-OCT) at each time point. The following data were recorded: types of drugs injected to the patient at baseline, number of previous injections, average injection interval before and after treatment delay, delay time from the scheduled date, period from the date of the first diagnosis of nAMD to the date of the delayed injection, and period between the baseline and the date of the delayed injection. The treatment regimen before and after the delayed injection was recorded as treat and extend (T&E) or as needed (pro re nata (PRN)) [18].

SS-OCT was performed on the enrolled eyes of all included patients using a swept-source DRI OCT Triton Plus (Topcon Co., Tokyo, Japan) at each follow-up point. Using the OCT scans, we recorded the central subfield thickness (CST) and any anatomical abnormalities including accumulated levels of subretinal fluid (SRF), intraretinal fluid (IRF), and subretinal hyperreflective material (SHRM) and presence of pigment epithelial detachment (PED), and measured the maximum height of the SRF and PED. CST, which refers to the average retinal thickness within a 1-mm-diameter ring based on the macula [19], was measured using built-in optical software (IMAGEnet 6 version 1.25.16650; Topcon Co., Tokyo, Japan). Anatomical abnormalities in OCT images, such as the levels of SHRM, SRF, PED, and IRF, were found to indicate exudative disease activity [20,21,22]. For patients with confirmed SRF accumulation or PED, the maximum SRF height (distance between the outer retina and the hyperreflective line of the RPE) was manually measured using the caliper tool in IMAGEnet software; a similar process was repeated to measure the maximum PED height (distance between the inner surface of the Bruch membrane and the outer surface of the RPE) by reviewing the OCT image [23,24] (Figure 1). 

All images and findings were reviewed and recorded by one researcher.

### Statistical Analysis

Statistical analyses were performed using IBM SPSS Statistics 25.0.0 (IBM Co., Armonk, NY, USA). Age, sex, laterality, number of previous injections, period of time from the last follow-up and appointment date to the date of delayed injection, the injection interval and the treatment regimen before and after the delay, and the type of drug used before the delay were analyzed as baseline characteristics. The changes in the injection interval before and after the delay were compared using the paired t-test, and the changes in the treatment regimen were compared using McNemar’s test. The average BCVA, CST, and maximum SRF and PED height at baseline, on the day of delayed injection, and 2, 4, and 6 months later were calculated and compared using a paired t-test. The presence of OCT findings during the follow-up period was recorded and statistically analyzed using McNemar’s test to determine if there were any significant changes compared to the baseline. Multiple regression analysis was used to evaluate the association of changes in BCVA (dependent variable) and maximum SRF height (dependent variable) at baseline and at 6 months’ follow-up, with other clinical and demographic factors (independent variables). Statistical significance was set at *p* ≤ 0.05.

## 3. Results

Fifty-seven eyes of fifty-seven patients were enrolled in the study. The mean age of patients on the day of delayed intravitreal anti-VEGF injection was 71.6 ± 7.6 years (range, 57–85 years). Of the patients, 42 were males and 15 were females. The right eye of 38 patients and the left eye of 19 patients underwent the treatment. An average of 14.7 ± 10.3 injections (range, 3–51) were administered to the enrolled eye before the date of delayed injection. The intervals from the first diagnosis to the delayed injection, from the last follow-up to the delayed injection, and from the scheduled visit date to the delayed injection (delayed time) were 46.23 ± 34.02 months (range, 6–164 months), 3.32 ± 1.21 months (range, 1.6–9.8 months), and 1.46 ± 1.24 months (range, 0.4–0.9 months), respectively. The average injection interval before and after the delayed injection date were 3.05 ± 1.45 months (range, 1–8.6 months) and 2.41 ± 1.46 months (range, 1–11.5 months), respectively. After the day of delayed injection, the injection interval decreased significantly (*p* = 0.002, paired *t*-test). At baseline, 36 patients were treated with the T&E regimen, and the rest with the PRN regimen. After the delayed injection, 48 patients were treated with the T&E regimen, and there was a significant increase in the number of patients treated with the T&E regimen (*p* < 0.001, McNemar’s test). There were 23 patients who had previously received a bevacizumab injection, 45 received an aflibercept injection, and 29 received a ranibizumab injection at baseline. The baseline characteristics of the patients are shown in Table 1.

Following SS-OCT, SRF, PED, SHRM, and IRF were identified at baseline in 33 (57.9%), 50 (87.7%), 11 (19.3%), and 9 (15.8%) of 57 patients, respectively. On the day of delayed injection, the number of patients with confirmed SRF accumulation increased to 44 of 57 (77.2%), which was significantly higher than that at baseline (*p* = 0.027, McNemar’s test). Two months later, there was no statistically significant change in OCT findings compared to baseline. Four months later, SHRM was found in 17 out of 55 patients (30.9%), which was significantly higher compared to baseline (*p* = 0.016, McNemar’s test). Finally, 6 months later, SRF was found in 43 (76.8%) and SHRM in 18 (32.1%) of 56 patients, and this increase from baseline was statistically significant (*p* = 0.021 and 0.016, respectively, McNemar’s test) (Table 2 and Figure 2).

Compared to baseline (0.29 ± 0.29), the mean BCVA continued to worsen significantly on the day of delayed injection (0.34 ± 0.31), 2 months later (0.36 ± 0.32), and 6 months later (0.38 ± 0.31) (logMAR, *p* = 0.044, 0.038, and 0.001, respectively, paired *t*-test). In the fourth month, the mean BCVA was lower than baseline (0.33 ± 0.33), but compared to the second month, it seemed to have improved; however, the difference was not statistically significant (logMAR, *p* = 0.141, paired *t*-test) (Table 3 and Figure 3a).

The CST increased significantly from baseline (227.82 ± 62.46) to the day of delayed injection (267.26 ± 77.74) (µm, *p* < 0.001, paired *t*-test), but there was no significant difference in CST changes thereafter. The maximum SRF height significantly increased from baseline (84.32 ± 89.33) to the day of delayed injection (147.51 ± 113.94) and 6 months later (121.38 ± 103.36) (µm, *p* < 0.001 and *p* = 0.027, respectively, paired *t*-test). The maximum PED height did not show any statistically significant changes during the follow-up period (Table 4 and Figure 3b). Consequently, the BCVA and maximum SRF height deteriorated significantly at 6 months’ follow-up compared to baseline.

The results of multiple regression analysis of possible factors influencing prognosis are summarized in Table 4 and Table 5. When analyzing the factors influencing the changes in BCVA and maximum SRF height, which were significantly worse at 6 months’ follow-up than at baseline, we found that the injection delay time or injection interval did not affect the deterioration, but that the high maximum SRF height was negatively correlated with the change in maximum SRF height (standardized beta coefficient = −0.716, *p* < 0.001, multiple regression analysis). That is, the higher the maximum SRF height at baseline, the lower the degree of SRF height deterioration 6 months later. We also found that the presence of baseline SRF was negatively correlated with the deterioration of maximum SRF height (standardized beta coefficient = −0.351, *p* = 0.015, multiple regression analysis). Thus, in patients who did not have SRF accumulation at baseline, the degree of deterioration of the maximum SRF height was severe at 6 months’ follow-up compared to those who developed SRF at baseline.

## 4. Discussion

In this study, nAMD patients whose intravitreal anti-VEGF injection was delayed due to the COVID-19 pandemic were observed for up to 6 months after the date of delayed injection. Changes in BCVA and CST, changes in maximum SRF and PED height, and anatomical changes that can be identified with serial OCT scans were examined and compared with data at the last follow-up prior to the date of delayed injection (baseline). The results showed that the injection interval significantly decreased from baseline to the date of delayed injection, and the treatment regimen was changed from PRN to T&E. Compared to baseline, SRF and SHRM levels increased 6 months later, whereas the BCVA and maximum SRF height significantly deteriorated. However, the CST and maximum PED height did not show any significant changes 6 months later compared to baseline. Overall, patients with delayed injections due to the COVID-19 situation had deteriorated clinical outcomes at 6 months after the delayed injections compared to baseline findings. We found that the presence and amount of SRF at baseline influenced the degree of deterioration of the maximum SRF height 6 months later.

With intravitreal anti-VEGF injection as a strategy for treating nAMD, monthly injections for up to 24 months were initially attempted, but other treatment strategies were devised because patients had difficulty meeting the monthly treatment schedule. In a PRN regimen, the injections are only administered when active neovascularization is present, whereas in a T&E regimen, the procedure interval is extended until the macular fluid completely disappears. If macular fluid persists, the treatment interval is usually shortened. The purpose of the T&E regimen is to identify the appropriate treatment interval that can stabilize the patient’s vision while suppressing disease activity [25]. The T&E regimen is a variant of the PRN regimen whereby injections are resumed if recurrence is detected, and then delivered with increasing intervals [26]. Particularly, patients with poor treatment outcomes are often switched from PRN to T&E, and a study has shown that BCVA was also improved in these patients [27]. In this study, after the injection was delayed due to COVID-19, the treatment strategy was switched from PRN to T&E for most patients, which reflects the short-term worsening of the disease due to delayed treatment.

In anti-VEGF therapy, it is imperative to maintain an adequate injection interval to control disease activity [28,29]. Some studies evaluating the changes in intravitreal injection therapy during the COVID-19 pandemic have consistently shown that intravitreal injection was not administered on time during this period [12,13,14]. Yang et al. [14] reported that the BCVA of patients was worse during the pandemic period than before, and Sevik et al. [12] found that BCVA and OCT disease activity worsened in patients with a delayed injection after the end of the COVID-19 lockdown period. Although some studies have investigated nAMD patients with delayed treatment due to COVID-19, to our knowledge, this is the first study to evaluate the clinical prognosis, including the quantitative analysis of anatomical abnormalities, by tracking nAMD patients with delayed intravitreal anti-VEGF injections due to COVID-19 for up to 6 months after the delayed injection date. Borrelli et al. [30] tracked nAMD patients up to two consecutive follow-ups after the COVID-19 pandemic to evaluate the short-term changes in the clinical course of nAMD patients due to the spread of COVID-19. They found that the follow-up interval increased due to COVID-19, the BCVA worsened, and the longer the visit time was delayed, the worse the BCVA. Similarly, in this study, BCVA significantly worsened after the COVID-19 outbreak compared to before the outbreak, but no association was observed between the injection delay time and the degree of BCVA deterioration. Elfalah et al. and Naravane et al. [15,16] compared the BCVA and CST before and after treatment delay in patients of various disease groups whose intravitreal injection was delayed due to COVID-19. In the case of nAMD patients, the BCVA worsened after a treatment delay in both studies, but CST decreased in one of the studies [15] and increased in the other study [16]. This difference between the two studies and the current study can be attributed to the fact that the two studies only analyzed a single time after the treatment delay. In the study by Yeter et al. [8], which is most similar to our study, the authors analyzed the changes in BCVA, CST, and OCT findings by following, up to an average of 3.5 months, nAMD patients with delayed intravitreal anti-VEGF injection. The authors reported that the injection interval shortened after the delay, and the CST and SRF, IRF, and SHRM levels increased initially, but decreased as the treatment started again, but the BCVA did not recover as much as before the treatment delay. Similarly, in our study, the injection interval decreased after the delay in injection, and the OCT findings deteriorated at the time of the injection delay, but improved after treatment resumed. Additionally, at a follow-up of 6 months after the treatment delay, the visual acuity did not recover as much as the baseline. However, in our study, the BCVA and OCT findings temporarily recovered and then worsened again during the 6-month period; particularly, the amount of SRF increased significantly (Figure 4).

Therefore, we can conclude that if the intravitreal injection is delayed, the clinical and anatomical symptoms may appear to improve within a short period of time after the delay, but may eventually worsen compared to the baseline. According to the study by Muether et al. [10], which showed similar results to ours, this may be explained by the long-lasting damage to the photoreceptors and retinal pigment epithelium that occurred during the injection delay affecting the visual prognosis more negatively. When treatment is resumed, a temporary functional and morphologic response is observed; however, it is not possible to overcome the damage accumulated at the time when treatment is delayed. However, a different pattern may show with a longer follow-up period; therefore, such patients will require a continuous long-term follow-up and close observation.

Since this study did not include a control group, we compared our results with those of several studies regarding the clinical course of nAMD patients who received timely intravitreal anti-VEGF treatment. In this study, the average period from diagnosis to delayed injection was 46.23 months, and the average period from the last follow-up to delayed injection was 3.32 months. Therefore, we reviewed changes in visual acuity over 9 months in nAMD patients at approximately 4 years after the initial treatment in several other studies. Singer et al. [31] studied nAMD patients treated with ranibizumab for >4 years. According to their study, the average BCVA decreased slightly between 45 and 54 months after treatment, but did not decrease by >3 words using the Early Treatment Diabetic Retinopathy Study (ETDRS) chart. In our study, the BCVA deteriorated from an average of 0.29 to 0.38 logMAR from baseline to 9 months, and there was a decrease of about 5 words when the results were converted into ETDRS words [32]. Gillies et al. [33] observed treatment-naive nAMD patients for up to 7 years after the start of intravitreal anti-VEGF injections. Similarly, they found out that the average BCVA decreased slightly from 48 to 60 months after the treatment started, but did not decrease by >3 words, and the degree of deterioration was more severe in our study. Other studies also showed similar results [34,35], wherein patients who had been treated for approximately 4 years did not show a decrease in BCVA by >3 words within 9 months of continued anti-VEFG treatment. Therefore, it can be considered that the BCVA decreased relatively more rapidly in our study.

In terms of SRF levels, in nAMD patients treated with ranibizumab or bevacizumab for up to 2 years, the change in the proportion of patients with SRF accumulation from 15 to 24 months after treatment was <10%, according to one study [36]. However, in our study, it increased by 18.9% over 9 months from baseline. In the previous study, the change in SRF height from 15 to 24 months after treatment was <10 μm [36], but in our study, it increased to 37.06 μm in 9 months. When these patients were observed for up to 5 years, it was shown that the SRF thickness decreased in the 5th year compared to the 2nd year [37]. Therefore, it would be difficult to see that patients who have been treated promptly for an average of ≥4 years will have a significant increase in SRF thickness within 9 months, as in the result of our study. Overall, as the intravitreal anti-VEGF therapy of patients in our study was delayed due to COVID-19, it can be considered that the BCVA and SRF amount of these patients worsened compared to those of patients who would undergo regular treatment.

In this study, patients with SRF accumulation were compared to patients without SRF accumulation at baseline, and those with a higher baseline maximum SRF height showed less deterioration of maximum SRF height at 6 months after follow-up (Table 4 and Table 5). The reason for this has not been clarified, but the following has been presumed. First, the injection interval before and after the delayed injection was shorter in patients with baseline SRF accumulation than in patients without any baseline increase in the SRF amount. In the case of patients with baseline SRF accumulation, it is highly likely that these patients had tolerable levels of SRF despite being continuously treated with intravitreal anti-VEGF injection [38]. Alternatively, the reason may be poor or no response to intravitreal anti-VEGF injections [39]. In this case, the injection drug can be changed, the treatment regimen can be switched to T&E, or the injection interval can be shortened, all while ensuring frequent follow-up observations [27,39]. In this study, the average injection interval of patients with SRF accumulation at baseline was 2.69 ± 1.17 months, and that of patients without SRF accumulation was 3.53 ± 1.67 months, and the injection interval of patients with baseline SRF accumulation was significantly shorter (*p* = 0.03, independent *t*-test). A similar trend was shown even after the treatment delay, and the injection interval was significantly shorter in patients with baseline SRF accumulation (1.99 ± 0.67 months in patients with baseline SRF accumulation, and 3.03 ± 2.01 months in patients without baseline SRF accumulation; *p* = 0.008, independent *t*-test). Therefore, it is possible that the short injection interval of patients with baseline SRF accumulation caused the increase in SRF levels to be relatively slow. Second, in patients with baseline SRF accumulation, although not statistically significant, the maximum SRF height at baseline was negatively correlated with the injection interval before and after the treatment delay (R = −0.262 before delayed injection and R = −0.343 after delayed injection; *p* = 0.140 and 0.051, respectively, Pearson’s correlation analysis). In other words, the higher the baseline maximum SRF height, the more severe the disease activity, and thus, the shorter the injection interval. Therefore, the shorter injection interval possibly had the effect of slowing down the deterioration of SRF height even after the injection was delayed due to COVID-19.

Our study has several limitations. First, its retrospective nature may cause selective bias, and because this study relied on medical records, and the intervals of follow-up of patients were different, in the case of patients who had a longer follow-up period or did not visit the scheduled treatment regularly, follow-up observations comparing all time points were unsuccessful. In particular, only 40 out of 57 patients visited at 2 months’ follow-up after the time of delayed injection. Additionally, this study was conducted on patients who visited a single tertiary medical center in Daegu, Korea, so the findings will be difficult to apply to the general population. Second, since in this study, we investigated progress in a short period of 6 months after the delayed injection, the prognosis for a long-term follow-up, such as for 1 year, was not discussed. In this short-term follow-up, we confirmed that the BCVA and SRF amount worsened compared to the baseline, but there remains insufficient information on whether the delay in injection affects the clinical course of the disease even after a follow-up of >1 year. Third, we confirmed that the BCVA deteriorated at 6 months of follow-up, but despite adjusting for other dependent variables through multiple regression analysis, we could not identify the factors that influenced the deterioration. The SRF was also confirmed to increase in frequency and amount at 6 months from baseline, and no factors influencing this were identified. Finally, as discussed above, this study lacks a control group of patients who had been treated for nAMD as scheduled, for comparison. Therefore, a prospective long-term study with a larger sample size is required.

## 5. Conclusions

In conclusion, when intravitreal anti-VEGF injections in nAMD patients were delayed due to the COVID-19 pandemic, the BCVA and SRF worsened 6 months after the treatment delay, and the presence of baseline SRF was found to reduce the degree of SRF height deterioration. Furthermore, after the injection was delayed, the disease activity seemed to improve in the first 2–4 months after the treatment was resumed, but it eventually worsened again after 6 months. This implies that the clinical prognosis of nAMD patients could worsen after a short-term follow-up of 6 months, and according to the results of this study, even if a similar global crisis occurs, patients with nAMD must still be prompted to attend scheduled visits and treatments on time.

## Figures and Tables

**Figure 1 jcm-11-02321-f001:**
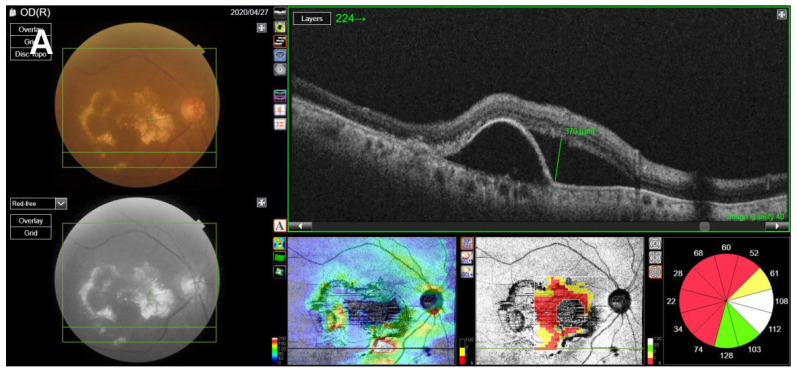
Measurement of SRF, PED, SHRM, and IRF. (**A**) The maximum SRF height was measured as the maximum distance between the outer retina and the hyperreflective line of the retinal pigment epithelium on the OCT image, using the caliper tool using IMAGEnet software. (**B**) The maximum PED height was measured as the maximum distance between the inner surface of the Bruch membrane and the outer surface of the retinal pigment epithelium. (**C**) Presence of SHRM, which is a morphological feature seen on OCT as hyperreflective material located external to the retina and internal to the retinal pigment epithelium. (**D**) Presence of IRF, which appears as dark cystic accumulations of fluid above the outer plexiform layer. SRF, subretinal fluid; PED, pigment epithelial detachment; SHRM, subretinal hyperreflective material; IRF, intraretinal fluid; OCT, optical coherence tomography.

**Figure 2 jcm-11-02321-f002:**
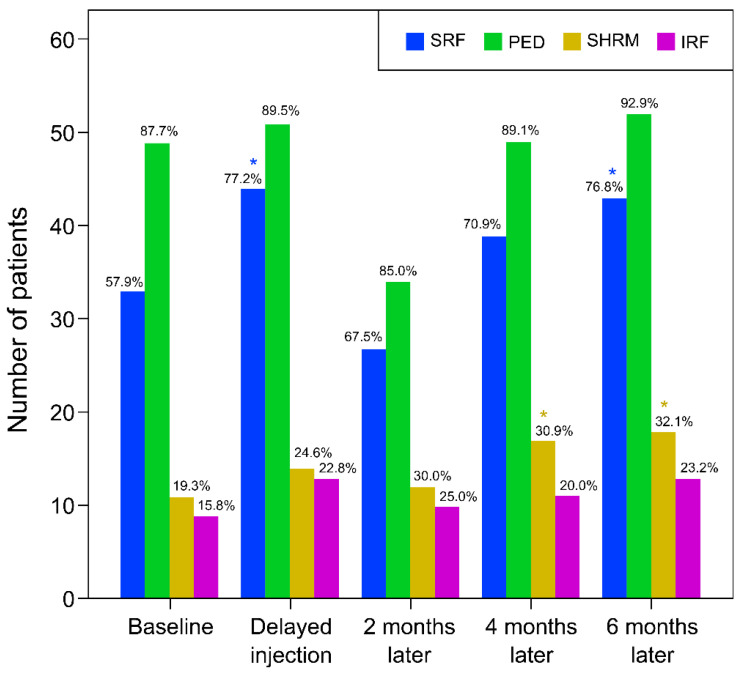
Bar chart displaying changes in structural OCT findings during the follow-up period. The height of the bars indicates the number of patients who showed the corresponding OCT finding. The percentage shown above the bar represents the percentage of patients who had the finding among the patients observed at that time. The proportion of patients with OCT findings of SRF, PED, and IRF increased after the time of delayed injection, and when the treatment was restarted, and the figure decreased 2 months after the treatment delay, but eventually gradually increased 4 and 6 months after the delay. The proportion of patients with SHRM gradually increased over time from baseline. OCT, optical coherence tomography; SRF, subretinal fluid; PED, pigment epithelial detachment; SHRM, subretinal hyperreflective material; IRF, intraretinal fluid. * Statistically significant compared to baseline.

**Figure 3 jcm-11-02321-f003:**
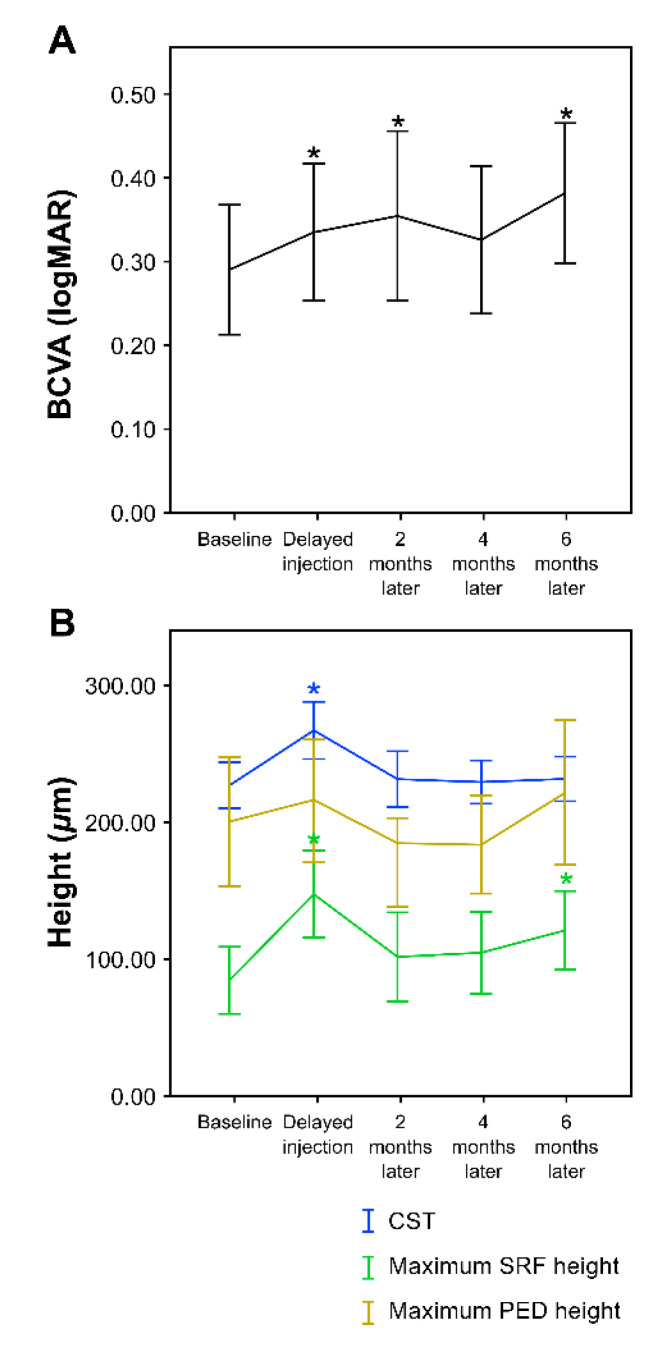
Line graph showing differences in BCVA, CST, maximum SRF height, and maximum PED height between visits. (**A**) Compared to baseline, the BCVA significantly worsened up to 2 months after the time of delayed injection. A trend of temporarily recovering BCVA was observed 4 months after the treatment delay, but it was not statistically significant. In the last follow-up (6 months after the delay), the BCVA was significantly worse than the baseline finding. (**B**) When the OCT findings were quantitatively analyzed, the overall, CST, maximum SRF height, and maximum PED height all deteriorated at the time of delayed injection and then recovered, but eventually showed a pattern of worsening again at approximately 6 months after the treatment delay. At the time of delayed injection, the CST and maximum SRF height showed statistically significant deterioration compared to baseline, and in the final follow-up (6 months), the maximum SRF height was statistically significantly worse than the baseline value. The maximum PED height showed no statistically significant change compared to baseline. Values are presented as means ±95% confidence interval. BCVA, best-corrected visual acuity; logMAR, logarithm of the minimum angle of resolution; CST, central subfield thickness; SRF, subretinal fluid; PED, pigment epithelial detachment. * Statistically significant compared to baseline.

**Figure 4 jcm-11-02321-f004:**
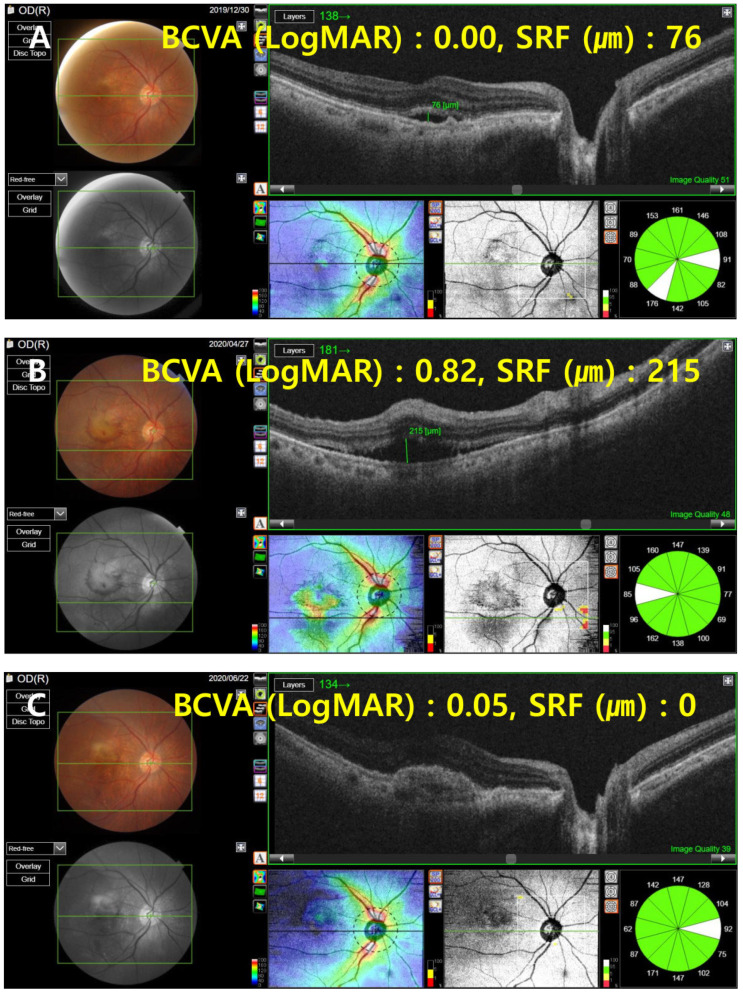
OCT change in a patient with neovascular age-related macular degeneration. (**A**) Baseline OCT image. (**B**) OCT image at the time of delayed injection. (**C**) Two months after the delayed injection. (**D**) Four months after the delayed injection. (**E**) Six months after the delayed injection. At the time of delayed injection, the BCVA was worse than the baseline, and the maximum SRF height (green line) was higher. After resumption of treatment, the BCVA and SRF height gradually improved, but eventually did not recover to baseline levels after 6 months after the treatment delay. OCT, optical coherence tomography; BCVA, best-corrected visual acuity; logMAR, logarithm of the minimum angle of resolution; SRF, subretinal fluid.

**Table 1 jcm-11-02321-t001:** Baseline characteristics of patients.

Characteristics		Number or Interval (Range)
Total number of eyes enrolled (patients)		57
Age (years)		71.6 ± 7.6 (57–85)
Sex (male-to-female)		42:15
Laterality (right-to-left)		38:19
Previous anti-VEGF injection (number of times)		14.7 ± 10.3 (3–51)
Delayed injection interval (months)		1.46 ± 1.24 (0.4–9)
Interval from diagnosis to delayed injection (months)		46.23 ± 34.02 (6–164)
Interval from baseline to delayed injection (months)		3.32 ± 1.21 (1.6–9.8)
Injection interval (months, before delayed injection vs. after delayed injection)		3.05 ± 1.45 (1–8.6) vs. 2.41 ± 1.46, (1–11.5), *p* = 0.002 ^a^ *
Treatment regimen at baseline (PRN-to-T&E)		21:36
Treatment regimen after delayed injection (PRN-to-T&E)		9:48, *p* < 0.001 ^b^ *
Type of injection drug (including duplicates) ^#^	Bevacizumab	23
	Aflibercept	45
	Ranibizumab	29

Values are presented as means ± standard deviation or as numbers. VEGF, vascular endothelial growth factor; PRN, pro re nata (as necessary); T&E, treat and extend. ^#^ Type of injections that patients received at least once before the time of delayed injection. ^a^ Paired t-test; ^b^ McNemar’s test, with comparison to the treatment regimen at baseline. * *p* < 0.05.

**Table 2 jcm-11-02321-t002:** Differences in OCT findings at different visits.

	Baseline (N = 57)	Delayed Injection (N = 57)	2 Months Later (N = 40)	4 Months Later (N = 55)	6 Months Later (N = 56)
OCT evidence of SRF, *n* (%)	33 (57.9)	44 (77.2), *p* = 0.027 ^a^ *	27 (67.5), *p* = 0.549 ^a^	39 (70.9), *p* = 0.096 ^a^	43 (76.8), *p* = 0.021 ^a^ *
OCT evidence of PED, *n* (%)	50 (87.7)	51 (89.5), *p* = 1.000 ^a^	34 (85.0), *p* = 1.000 ^a^	49 (89.1), *p* = 1.000 ^a^	52 (92.9), *p =* 0.500 ^a^
OCT evidence of SHRM, *n* (%)	11 (19.3)	14 (24.6), *p* = 0.250 ^a^	12 (30.0), *p* = 0.250 ^a^	17 (30.9), *p* = 0.016 ^a^ *	18 (32.1), *p* = 0.016 ^a^ *
OCT evidence of IRF, *n* (%)	9 (15.8)	13 (22.8), *p* = 0.125 ^a^	10 (25.0), *p* = 0.375 ^a^	11 (20.0), *p* = 0.625 ^a^	13 (23.2), *p* = 0.219 ^a^

Values are presented as numbers (percentages). OCT, optical coherence tomography; SRF, subretinal fluid; PED, pigment epithelial detachment; SHRM, subretinal hyperreflective material; IRF, intraretinal fluid. ^a^ Comparison with baseline, using McNemar’s test. * *p* < 0.05.

**Table 3 jcm-11-02321-t003:** Changes in BCVA and OCT measurements during the follow-up period.

	Baseline (N = 57)	Delayed Injection (N = 57)	2 Months Later (N = 40)	4 Months Later (N = 55)	6 Months Later (N = 56)
BCVA (by logMAR)	0.29 ± 0.29	0.34 ± 0.31, *p* = 0.044 ^a^ *	0.36 ± 0.32, *p* = 0.038 ^a^ *	0.33 ± 0.33, *p* = 0.141 ^a^	0.38 ± 0.31, *p* = 0.001 ^a^ *
CST (µm)	227.82 ± 62.46	267.26 ± 77.74, *p* < 0.001 ^a^ *	231.83 ± 64.52, *p* = 0.708 ^a^	229.47 ± 58.16, *p* = 0.926 ^a^	231.80 ± 61.15, *p* = 0.757 ^a^
Maximum SRF height (µm)	84.32 ± 89.33	147.51 ± 113.94, *p* < 0.001 ^a^ *	101.64 ± 101.20, *p* = 0.677 ^a^	104.41 ± 107.50, *p* = 0.260 ^a^	121.38 ± 103.36, *p* = 0.027 ^a^ *
Maximum PED height (µm)	200.42 ± 169.98	215.71 ± 161.11, *p* = 0.347 ^a^	184.05 ± 141.78, *p* = 0.286 ^a^	184.10 ± 126.56, *p* = 0.373 ^a^	222.00 ± 190.64, *p* = 0.292 ^a^

Values are presented as means ± standard deviation or as numbers. The maximum PED and SRF heights were recorded as 0 µm on the day it disappeared if they were present before, but later disappeared. In addition, in the case of new PED and SRF findings, all prior parameters were calculated as 0 µm. BCVA, best-corrected visual acuity; logMAR, logarithm of the minimum angle of resolution; CST, central subfield thickness; PED, pigment epithelial detachment; SRF, subretinal fluid. ^a^ comparison versus baseline, paired *t*-test. * *p* < 0.05.

**Table 4 jcm-11-02321-t004:** Results of multiple regression analysis of the association of changes in BCVA and maximum SRF height at baseline and 6 months’ follow-up (final) with other variables.

	Change in BCVA		Change in Maximum SRF Height	
	Standardized Beta Coefficient (SE)	*p*-Value	Standardized Beta Coefficient (SE)	*p*-Value
Age (years)	−0.017	0.918	−0.038	0.764
Sex	−0.106	0.499	0.135	0.274
Previous anti-VEGF injection (number of times)	−0.361	0.107	−0.238	0.171
Delayed injection time (months)	−0.431	0.187	−0.146	0.562
Injection interval before delayed injected (months)	−0.081	0.679	−0.246	0.111
Injection interval after delayed injected (months)	−0.238	0.201	−0.181	0.212
Interval from diagnosis to delayed injection (months)	0.197	0.356	0.132	0.428
Interval from last follow-up to delayed injection (months)	0.428	0.201	−0.089	0.731
Baseline BCVA (logMAR)	−0.293	0.073	−0.064	0.609
Baseline CST (µm)	−0.106	0.544	−0.264	0.059
Baseline maximum SRF height (µm)	0.318	0.184	−0.716	<0.001 *
Baseline maximum PED height (µm)	−0.282	0.138	0.238	0.110

BCVA, best-corrected visual acuity; SRF, subretinal fluid; SE, standard error; CST, central subfield thickness; VEGF, vascular endothelial growth factor; logMAR, logarithm of the minimum angle of resolution; PED, pigment epithelial detachment. * *p* < 0.05.

**Table 5 jcm-11-02321-t005:** Results of multiple regression analysis of the association of changes in BCVA and maximum SRF height at baseline and 6 months’ follow-up (final) with baseline OCT findings.

	Change in BCVA		Change in Maximum SRF Height	
	Standardized Beta Coefficient (SE)	*p*-Value	Standardized Beta Coefficient (SE)	*p*-Value
SRF+	0.158	0.274	−0.351	0.015 *
PED+	0.179	0.230	0.058	0.678
SHRM+	0.150	0.304	−0.104	0.458
IRF+	−0.017	0.907	−0.168	0.220

BCVA, best-corrected visual acuity; OCT, optical coherence tomography; SRF, subretinal fluid; PED, pigment epithelial detachment; SHRM, subretinal hyperreflective material; IRF, intraretinal fluid. * *p* < 0.05.

## Data Availability

The data that support the findings of this study are available on request from the corresponding author. The data are not publicly available due to privacy or ethical restrictions.

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
