# Peer review of "Impact of Delayed Intravitreal Anti-Vascular Endothelial Growth Factor (VEGF) Therapy Due to the Coronavirus Disease Pandemic on the Prognosis of Patients with Neovascular Age-Related Macular Degeneration"

_jcm, 2022, doi:10.3390/jcm11092321_

Round 1
Reviewer 1 Report
The paper is interesting and well written. This issue (delay of treatment in the pandemic era) has already been addressed by other authors, but suggestions about how to fix this problem are welcome.
In the abstract:
"The study estimated the outcome of delayed intravitreal anti-vascular endothelial growth factor (VEGF) therapy due to the coronavirus disease (COVID-19) pandemic..." --> please put COVID-19 before disease
"Delayed intravitreal anti-VEGF therapy was significantly associated with BCVA and SRF changes in nAMD patients"--> please specify clearly what "vhanges" you mean. I would suggest: "Delayed intravitreal anti-VEGF therapy caused by coronavirus pandemic was significantly associated with a worse BCVA and residual SRF in nAMD patients".
At line 39:
“Injecting drugs such as aflibercept, ranibizumab, and bevacizumab into the vitreous cavity, which block abnormal growth of the retinal and choroidal vessels, is currently the main treatment for nAMD”à this sentence sounds better as “Injecting anti VEGF drugs such as aflibercept, ranibizumab, and bevacizumab into the vitreous cavity, is currently the gold standard” for nAMD.
Please specify in the text which was the average delay between nAMD diagnosis and injection BEFORE the pandemic, in the authors’ hospital.
Reviewer 2 Report
Thank you for providing an opportunity to review the manuscript. Authors of the manuscript have conducted an interesting study to evaluate the impact of delayed intravitreal anti-vascular endothelial growth factor (VEGF) therapy due to the coronavirus disease pandemic on the prognosis of patients with neovascular age-related macular degeneration. The study is very interesting.
- Please add the conclusions to the abstract.
- Please provide a high level summary of any similar studies conducted (from discussion section) in the introduction section.
- If possible, please provide flowchart for the study design in the manuscript or provide a high level summary of the study design in the last paragraph of the introduction. Something like… “This study was a retrospective study aimed to……….”
- Please rephrase lines 39-40 as “Injecting drugs such as aflibercept, ranibizumab, and bevacizumab into the vitreous cavity is currently the main treatment for nAMD. These agents block abnormal growth of the retinal and choroidal vessels.” Please avoid long sentences in the manuscript.
- Please provide the inclusion and exclusion criteria as a supplementary information and include only the key inclusion and exclusion criteria in the manuscript (presentation in a table format will be easy to read)
- Please remove the abbreviation of the researcher on line 136.
- Please delete lines 270 to 273
- Line 274: please change postponed to delayed
- Line 338-340: Can you please provide the rationale for short term improvement and worsening later?
- Please use the term “baseline” instead of “before the delay” in case the meaning is same.
